# Role of HBcAb Positivity in Increase of HIV-RNA Detectability after Switching to a Two-Drug Regimen Lamivudine-Based (2DR-3TC-Based) Treatment: Months 48 Results of a Multicenter Italian Cohort

**DOI:** 10.3390/v15010193

**Published:** 2023-01-10

**Authors:** Vincenzo Malagnino, Romina Salpini, Elisabetta Teti, Mirko Compagno, Ludovica Ferrari, Tiziana Mulas, Valentina Svicher, Marta Zordan, Monica Basso, Giuliana Battagin, Sandro Panese, Maria Cristina Rossi, Renzo Scaggiante, Daniela Zago, Marco Iannetta, Saverio Giuseppe Parisi, Massimo Andreoni, Loredana Sarmati

**Affiliations:** 1Clinical of Infectious Diseases, Tor Vergata Policlinic of Rome, 00133 Rome, Italy; 2Department of Medicine of Systems, Tor Vergata University of Rome, 00133 Rome, Italy; 3Department of Experimental Medicine, Tor Vergata University of Rome, 00133 Rome, Italy; 4Department of Biology, Tor Vergata University of Rome, 00133 Rome, Italy; 5Department of Molecular Medicina, University of Padua, 35100 Padova, Italy; 6UOC Malattie Infettive, Ospedale di Vicenza, 36100 Vicenza, Italy; 7UOC Malattie Infettive, Ospedale di Venezia, 30122 Venezia, Italy; 8UOC Malattie Infettive, Ospedale di Treviso, 31100 Treviso, Italy; 9UOC Malattie Infettive, Ospedale di Belluno, 32100 Belluno, Italy

**Keywords:** HIV/HBV, 2DR, HBcAb, hepatitis B virus (HBV), Occult Hepatitis B Infection (OBI)

## Abstract

The aim of this study was to evaluate whether the presence of anti-hepatitis B (HBV) c antibodies (HBcAb positivity) could influence the control of HIV viremia in patients living with HIV (PLWH) who switch to two-drug antiretroviral therapy (2DR) containing lamivudine (3TC) (2DR-3TC-based). A retrospective multicentre observational study was conducted on 160 PLWH switching to the 2DR-3TC-based regimen: 51 HBcAb-positive and 109 HBcAb-negative patients. The HBcAb-positive PLWH group demonstrated a significantly lower percentage of subjects with HIV viral suppression with target not detected (TND) at all time points after switching (24th month: 64.7% vs. 87.8%, *p* < 0.0001; 36th month 62.7% vs. 86.8%, *p* = 0.011; 48th month 57.2% vs. 86.1%, *p* = 0.021 of the HBcAb-positive and HBcAb-negative groups, respectively). Logistic regression analysis showed that the presence of HBcAb positivity (OR 7.46 [95% CI 2.35–14.77], *p* = 0.004) could favour the emergence of HIV viral rebound by nearly 54% during the entire study follow-up after switching to 2DR-3TC.

## 1. Introduction

Co-infection with the hepatitis B virus affects more than 7% of people living with HIV, with a prevalence of up to 16% in the highest HBV endemic areas such as West and Central Africa [1]. It represents an additional risk factor for progression to AIDS-defining pathology [2], lower immunological recovery [3], but also the development of severe liver disease and end-stage liver disease [4]. In addition to the condition of chronic infection (HBsAg positivity), a proportion of PLWH HBsAg-negative patients have antibodies against the HBV core antigen (HBcAb) as a sign of previous infection, and many of these may present HBV replication (HBV-DNA positivity) in the absence of antigen HBs, in a defined condition of Occult Hepatitis B Infection (OBI). Moreover, a recent update of the Taormina statement on the OBI condition defined the positivity for HBcAb as a surrogate marker of occult infection in the absence of data demonstrating HBV-DNA positivity in plasma and/or tissue [5]. Immunosuppressive conditions, driven by HIV infection or induced by malignancies or immunotherapies, can favor the presence of OBI. In this light, this condition is quite common in PLWH, particularly in patients with low CD4+ counts, that can experience a lack of immune control of HBV infection [6].

Recently, we demonstrated that HBcAb positivity was associated with a delay in achieving HIV undetectability and the onset of viral rebound in HBcAb-positive PLWH after triple ART initiation [7], suggesting the possible contribution of transient reactivation of HBV to poor HIV control. Actually, the presence of detectable HBV-DNA was demonstrated in 15% of ART-treated HIV/HBcAb-positive patients from an Italian cohort [8], and the presence of “cryptic” HBV viremia (below 20 IU/mL of commercial tests) was demonstrated in 29% of PLWH HBcAb-positive ART patients [9]. Furthermore, we created a multicentre Italian cohort of patients on Lamivudine-based-dual therapy (2DR-3TC-based) and verified the impact of HBcAb positivity, demonstrating that HbcAb+ was significantly associated with the worst HIV replicative control at all time points of follow-up and correlated with a nearly 3-fold increase in the risk of HIV-RNA detectability at 24 months after the 2DR-3TC switch (OR 2.7 [95% CI 1.05–6.9], *p* = 0.038) [10]. Here we present an update of the data 48 months after the 2DR-3TC-based switch.

## 2. Materials and Methods

### 2.1. Study Design

A retrospective observational study was conducted on 230 PLWH switched to the 2DR-3TC-based regimen and followed up thereafter. Patients were enrolled at different clinical sites: the Infectious Diseases Clinic of the Tor Vergata Polyclinic in Rome, the Department of Molecular Medicine of the University of Padua, the Units of Infectious Diseases of Vicenza, and the Venezia and Treviso hospitals. For this study, a database was built, including all patients’ data at HIV infection diagnosis (baseline), at the start of ART treatment, and at follow-up visits. In particular, the following data were collected for all patients: demographic information, HBV serology at baseline, CD4+ cell count at baseline and at the time of the 2DR-3TC switch, calendar year of HIV infection diagnosis, ART composition before the 2DR-3TC switch, and HIV-RNA viral load at 6, 12, 18, 24 months before the 2DR-3TC switch, at the time of the switch and at 6, 12, 18, 24, 36 and 48 months after.

### 2.2. Inclusion and Exclusion Criteria

As shown in Figure 1, of the 230 patients followed in the multicentre cohort switched in 2DR-3TC-based and HBsAg-negative, 1 patient was excluded because he started the 2DR-3TC regimen as naïve, 58 patients were excluded due to lack of HBV serological data, 2 patients were excluded because they were in a virologic rebound at the time of the switch, and 9 patients were excluded because lost to follow up (intended as the lack of data after the first 12 months after the switch) after the 2DR-3TC-based switch. Ultimately, one hundred sixty patients were studied.

### 2.3. Laboratory Testing for the Diagnosis of HBV and HIV Infections

HBV serological markers were measured using immune-enzymatic assays (Roche/Cobas Diagnostics, Rotkreuz, Switzerland). Plasma HIV-RNA levels were measured using a commercial test characterized by a lower limit of quantification (LLOQ) for HIV-RNA of 20 copies/mL COBAS AmpliPrep/COBAS TaqMan HIV-1 Test, v2.0). The test allows detection and also the presence of HIV-RNA < 20 HIV-RNA copies/mL, a condition that is defined as HIV-RNA detectability below the LLOQ of the assay.

### 2.4. HIV Viral Load Definitions

All patients enrolled in the study had an HIV viral load < 50 copies/mL at the time of the switch to 2DR-3TC treatment. Levels of HIV-RNA detectable below the limits of quantification of the commercial assay (20 cp/mL) have been found to be associated with an increased risk of virological failure, virological rebound, and the development of resistance [11,12]. To evaluate the HIV viremia kinetics after the 2DR switch in the study cohort, all HIV viral load measurements were carried out at 6, 12, and 24 months before and at 6, 12, 24, 36, and 48 months after the transition to the 2DR-3TC regimen were considered. Based on the results of HIV-RNA obtained by the specified commercial Real-Time assay, the following definitions were adopted: (1): viral load undetectable if viremia was less than 20 copies/mL, target not detected; (2) very low-level viremia if viremia was detected but at a level below the LLOQ of 20 copies/mL; and (3) viral load detectable if viremia was above 20 copies/mL. Viremia values higher than 20 copies/mL and the detection of HIV-RNA below the LLOQ of 20 copies/mL were considered signs of active HIV replication [13].

### 2.5. Endpoints

The endpoint of the study was to evaluate differences in the maintenance of HIV viremia suppression between the two PLWH populations of HBcAb-positive and HBcAb-negative patients 6, 12, and 24 months after the 2DR-3TC-based switch.

### 2.6. Statistical Methods

Data were collected, and the dataset was assembled using Excel 2019 version 16.3. All statistical analyses were conducted using STATA 14.2 (College Station, TX, USA), and graphs and confirmation analyses were performed using GraphPad Prism. 8.2.1.

The study population is described using proportions and percentages for categorical values, median measurements, and interquartile ranges (IQRs) for continuous values. The comparison between PLWH-HBcAb-positive and PLWH-HBcAb-negative patients was performed with the Kruskal-Wallis test for continuous variables and with the chi-squared test or Fisher’s exact test, when appropriate, for categorical variables. Univariable odds ratios (ORs) associated with a lack of HIV RNA undetectability at T24 and their 95% confidence intervals (CIs) were calculated using logistic regression. A multivariable model was constructed in which risk factors with *p* < 0.1 in univariable analysis were included in a full model and were then excluded in a backward-stepwise fashion if *p* < 0.05 using the likelihood ratio test.

### 2.7. Ethic Statements

The study protocol and related informed consent were submitted and approved by the Independent Ethics Committee at Policlinico Tor Vergata [Protocol Number 216/16, version 1.0]; the study was structured with respect to privacy. Personal information was treated in a confidential manner, and clinical data were anonymized in accordance with the Helsinki Declaration (version October 2013). All included patients signed an informed consent for inclusion in the observation.

## 3. Results

### 3.1. Description of the Study Population

The full description of the study population is shown in Table 1. One hundred and sixty patients switched to a 2DR-3TC regimen were studied, with median age was 42.7 years (IQR 32–54), prevalently males, and a median duration of HIV infection of approximately nine years. As regards the lymphocyte subpopulations, an effective immunological recovery after the initiation of triple ART was demonstrated CD4+ nadir median 287 cell/mm^3^, while the median CD4 cell count at the time of the switch was 753/mm^3^). Triple ART treatment before the switch to 2DR-3TC-based confirmed a balance in the choice of anchor drugs used: 48 patients (30.1%) were on 2 NRTI+ a protease inhibitor (PI), 51 (32.1%) with 2 NRTI+ non-nucleoside reverse transcriptase inhibitor (NNRTI), 47 (29.6%) with 2 NRTI+ an integrase inhibitor (INI) and 13 (8.2%) with other not conventional therapeutic regimens (four-drug therapy or dual therapy NRTI-sparing). A total of 22 patients experienced HIV virological rebound, but only six virological failures with the appearance of resistance-conferring mutations, including M184V (data not shown).

Importantly The vast majority of PLWH reached and showed the viral load undetectability or <20 cp/mL with a target detected 24 months before the 2DR-3TC-based switch (145 patients, 95.7%) and before 12 months after the start of ART (143 patients, 89.4%). At the time of the switch to 2DR treatment, 14 subjects (8.8%) showed a detectable HIV-RNA >20 cp/mL but <50 cp/mL, while 30 (18.7%) subjects showed a viremia value below 20 copies/mL with a target detected. Furthermore, 19 patients (11.9%) were HCVAb+, 10 were treated with Direct Antiviral Agents (DAAs) before 2DR-3TC-switch, and nine never resulted positive to HCV-RNA (data not shown). About one-third of the patients [51/160 (31.9%)] tested positive for HBcAb. HBV-DNA was not available for all patients, only 85 (53.1%) were tested at least once before the switch, and HBV-DNA resulted undetectable (data not shown).

### 3.2. HIV Viral Load after Switch to 2DR-3TC-Based

The 2DR simplification schemes were distributed as follows: 37 patients (23.2%) with 3TC + PI (Atazanavir [ATV] or Darunavir [DRV] regardless of whether boosted with ritonavir [RTV] or cobicistat [COBI]) and 123 (76.9%) with 3TC + Dolutegravir (DTG). All patients maintained virological suppression at the different observation times (6, 12 [data not shown], 24, 36, and 48 months after the switch), although with different levels of low-level viremia: at 24 months after simplification (data available for 141 patients), 112 (79.4%) resulted completely undetectable, while 23 (16.3%) showed a detectable HIV viremia below 20 cp/mL and six (4.3%) >20 cp/mL; at 36 months (data available for 127 patients) 119 (93.8%) had an HIV viremia less than 20 cp/mL (98 [77.2%] completely undetectable, and 21 [16.6%] detectable), while 27 patients (6.2%) showed residual HIV viremia at this observation time; at T48 (data available for 93 patients) 85 (91.4%) patients maintained HIV-RNA below 20 cp/mL, with 72 (77.4%) completely undetectable, while 13 (14%) detectable, and eight (8.6%) with virus detected higher than 20 cp/mL.

### 3.3. Comparison between HBcAb-Positive and HBcAb-Negative Patients

#### 3.3.1. Comparison at Time of 2DR Switch

As described above, of 160 patients included, 51 (31.9%) were HBcAb-positive, while 109 (68.1%) were HBcAb-negative. A comparison between HBcAb-negative and HBcAb-positive patients is reported in Table 2. HBcAb-positive patients tended to be older (HBcAb-positive 47 years [IQR 37–51], *p* = 0.0014) and with a long history of HIV infection (calendar year of HIV infection in HBcAb-positive 2007 while in HBcAb-negative 2010, *p* = 0.039), differently from our previous work, probably for the greater number of young patients undergoing 2DR-3TC based in the early phase, in a pre-emptive switch strategy. This feature also explains the absence of a difference in the choice of the third drug in the first-line triple ART strategies (*p* = 0.33). Conversely, immunological status described by CD4+ nadir and CD4+ cell count at the time of the 2DR-3TC switch was confirmed: the lower CD4+ nadir value, which tends to be lower in HbcAb-positive than negative patients (222/mm^3^ [IQR 69–335], *p* = 0.023), describes a more severe baseline HIV infection, but the CD4+ level measured at the time of switching to 2DR showed no difference between patients HbcAb-positive (683/mm^3^ [IQR 547–944]) and HbcAb-negative (680/mm^3^ [IQR 545–926]) (*p* = 0.91). Furthermore, in accordance with the previous study, no differences were found between the two subpopulations in the frequency of HIV viral rebound. In the pre-switch follow-up period and the HIV viral load measurements at 24 and 12 months before the switch and at the time of simplification to 2DR (*p* = 0.12; *p* = 0.41; *p* = 0.40; *p* = 0.87, respectively).

#### 3.3.2. Comparison after 2DR Switch

In the two subpopulations, 2DR containing Dolutegravir (DTG) was the most frequent choice (40 HBcAb-positive patients [784%] and 83 HBcAb-negative patients [76.1%], *p* = 0.74). Since the previous study described data 24 months after the switch, here only data starting from T24 will be described, confirming the difference in the maintenance of HIV-RNA virological suppression during post-2DR-3TC-based switch follow-up in HBcAb-positive patients.

At T24, from the switch to 2DR-3TC-based HbcAb-positive patients had less frequently achieved complete undetectability than HbcAb-negative patients (64.7% vs. 87.8%, *p* < 0.0001), data confirmed at T36 and T48. In fact, at T36 (available follow-up data for 127 patients), 32 HbcAb-positive patients (62.7%) had an undetectable HIV viremia lower than 20 cp/mL (and 15 [29.4%], <20 cp/mL with HIV-RNA detectable), while 66 patients HbcAb-negative (86.8%) a viral load completely undetectable (and six [7.9%] with detectable HIV viremia lower than 20 cp/mL) (*p* = 0.011). Likewise, at T48 (data available for 93 patients), 85.8% of HBcAb-positive patients maintained virological suppression with HIV-RNA below 20cp/mL (of these, 57.2% completely undetectable) compared to 93.8% of HBcAb-negative patients (of which 56 [86.1%] completely undetectable) (*p* = 0.021) (Results described in Figure 2). Patients completely undetectable in all time points of the study during follow-up after the switch to 2DR-3TC-based were 33 HbcAb-negative patients (50.7%) vs. one HbcAb-positive patient (3.5%).

To assess the impact on new therapeutic regimens with integrase inhibitors, the same analysis was performed only for patients on 2DR treatment containing DTG with confirmation of the disparity of virological control after the switch: in particular, patients switched in 3TC + DTG completely undetectable in all time points of the study during follow up after switch to 2DR-3TC-based were 22/83 (26.1%), HbcAb-negative patients, while no HbcAb-positive patient no HBCAb+ patient managed to consistently maintain complete virological suppression during the 48-month follow-up.

### 3.4. Risk Factor for Lack of Achievement of Viral Suppression 48 Months after 2DR-3TC-Based Switch

A logistic regression analysis (Table 3) was performed with the production of a multivariate model including Age, calendar year of HIV infection, CD4+ lymphocyte nadir (considered an increase of 250 cells/mm^3^), history of previous virological rebound and HBcAb-positivity. The analysis describes how positivity for HbcAb-positivity determines a risk of incurring poor control of HIV infection during the first 48 months from the switch to 2DR-3TC-based approximately over 50% more as compared to a negative patient (OR 0.46 [95% CI 0.05–0.77], *p* = 0.004).

## 4. Discussion

The study confirmed after 48 months from the 2DR-3TC-based switch in PLWH that HBcAb-positivity is associated with the worst HIV viremia control after 48 months of follow-up and less frequently maintains the persistent virological suppression during the entire follow-up. Moreover, we showed that HBcAb-positive PLWH has a 54% greater risk of experiencing detectable HIV viremia than reduced HIV viremia control for 48 months after the 2DR switch.

The study substantially confirms and extends the previous data, with the important difference in the study populations caused by the prolonged follow-up (48 months) and the introduction of new patients within our multicentre cohort. For example, in the previous work, the median of the calendar year of infection was 2009 (IQR 2003–2013), while in this study, it is 2013 (IQR 2008–2016) and also with younger age (42.5 in this study vs. 52.5). This also affects the percentage of simplified 2DR-3TC-based patients containing Dolutegravir (DTG) (123 [76.9%] in this study vs. 107 [64.5%] in the previous one), making the finding of poor virological control more current. HIV-RNA in the process of simplification of HBcAb-positive patients given the continuous increase of patients in simplification with 3TC + DTG, which now represent, according to a recent study by the European RESPOND cohort, about a quarter (22.4%) of the 2DRs in use in PLWH [14]. Fundamental to this goal were the GEMINI one and two studies [15]. Regarding HBV infection, the study protocols provide for the non-exclusion of Subjects positive for anti-HBc (negative HBsAg status) and positive for anti-HBs (past and/or current evidence) because they are considered immune to HBV. Our findings are supported by a recent study by Salpini et al. [9], showing that approximately half of 101 PLWH with anti-HBc positivity presented signs of HBV activity (HBV-DNA and/or HBV-RNA positivity) under an antiviral regimen containing tenofovir. Even more, this study also demonstrated that a relevant fraction (40%) of anti-HBc positive patients experienced a virological HBV reactivation [16] after TDF/TAF withdrawal. These findings can explain the correlation between anti-HBc positivity and the loss of HIV undetectability after the switch to a two-drug regimen containing lamivudine, which emerged in this study. Indeed, it can be hypothesized that the use of a less potent anti-HBV drug as lamivudine can induce the reactivation of HBV replication that, in turn, can lead to worse HIV control. This is also in line with data from the literature demonstrating how active coinfections can exacerbate HIV replication and disease progression [13]. Furthermore, the observation of occasional or persistent HIV viremia above the test threshold in patients on ART is common and has often been associated with virologic failure: Laprise et al. [17] described that persistent HIV-RNA detection below 50 copies/mL at 1, 2, and 5 years after ART initiation is associated with 5-, 10-, and 22-fold cumulative risks of virologic failure, respectively; however Maggiolo and colleagues [12] demonstrated that detection of extremely low levels of HIV-RNA (above 3 cp/mL) could be predictive of virological failure. It should be noted that no patient experienced virological failure with the appearance of resistance-associated mutations (RAM). In particular, no patient experienced two consecutive measurements greater than 50 cp/mL during follow-up or the detection of new AIDS-related events.

A previous study has demonstrated that the HBV protein known as HBx can increase the level of transcription of HIV genes, thus further enhancing HIV replication [18]. In this light, HBV reactivation observed after switching to a 2DR can favor the release of the HBx protein, thus jeopardizing the full control of HIV replication.

At the same time, persistent HBV replication (albeit at a low level) could have a detrimental impact on immunological recovery, thus further contributing to HIV rebound after switching to a 2DR [19,20].

Overall findings support a synergistic activity between HIV and HBV, thus highlighting the importance of setting up a proper screening of HBV markers to optimize the management of HIV-infected patients candidate to switch to a 2DR.

Regarding HBcAb-negative patients, compared to the previous study, the percentage of patients with a lack of virological suppression of HIV is reduced. Indeed, although only 33 HBcAb- patients (50.7%) consistently maintain virological suppression of HIV during the entire follow-up post-switch, nearly 90% of this subpopulation is TND at T24 (79 pts, 87%), T36 (66, 86.8%) and T48 (56, 86.1%), in line with WHO “90 90 90 strategy” [21].

This difference can be explained, as previously mentioned, with the inclusion over time of younger patients with more recent HIV infection and, therefore, less use of obsolete 2DR-3TC-based PIs-containing. In the previous study, Age was a recognized risk factor in the occurrence of virological rebound of HIV-RNA in the 24 months post-2DR-3TC-based-switch (OR 1.08 [95% CI 1.04–1.13], *p* < 0.0001 at multivariate analysis) and the use of 2DR-3TC_based containing PIs was described in 59 patients (35.5%) compared with 37 (23.2%) in the present study.

Finally, the study has some limitations, given the retrospective nature of the study. Evaluation in HBV-DNA viremia. In particular, the reason for the switch (pre-emptive switch, drug interaction, drug toxicity, or cost) is not homogeneous for all patients and is also linked to the various simplification schemes taken into consideration (PI or INI based). A further limit is the absence of data on adherence. The follow-up of patients in the present study might have been too short to observe failures. Finally, HCV serostatus and a risk factor for HIV transmission were not available for all patients included in the study.

On the basis of the data described, the role of previous HBV infection and potential OBI could be included among the risk factors for a viral rebound after the switch to the 2DR-3TC-based regimen. In addition, periodic screening of complete HBV serology, monitoring of HBV-DNA in HbcAb-positive patients with or without HbsAb, and analysis of liver fibrosis evolution in these patients should be strongly encouraged.

## Figures and Tables

**Figure 1 viruses-15-00193-f001:**
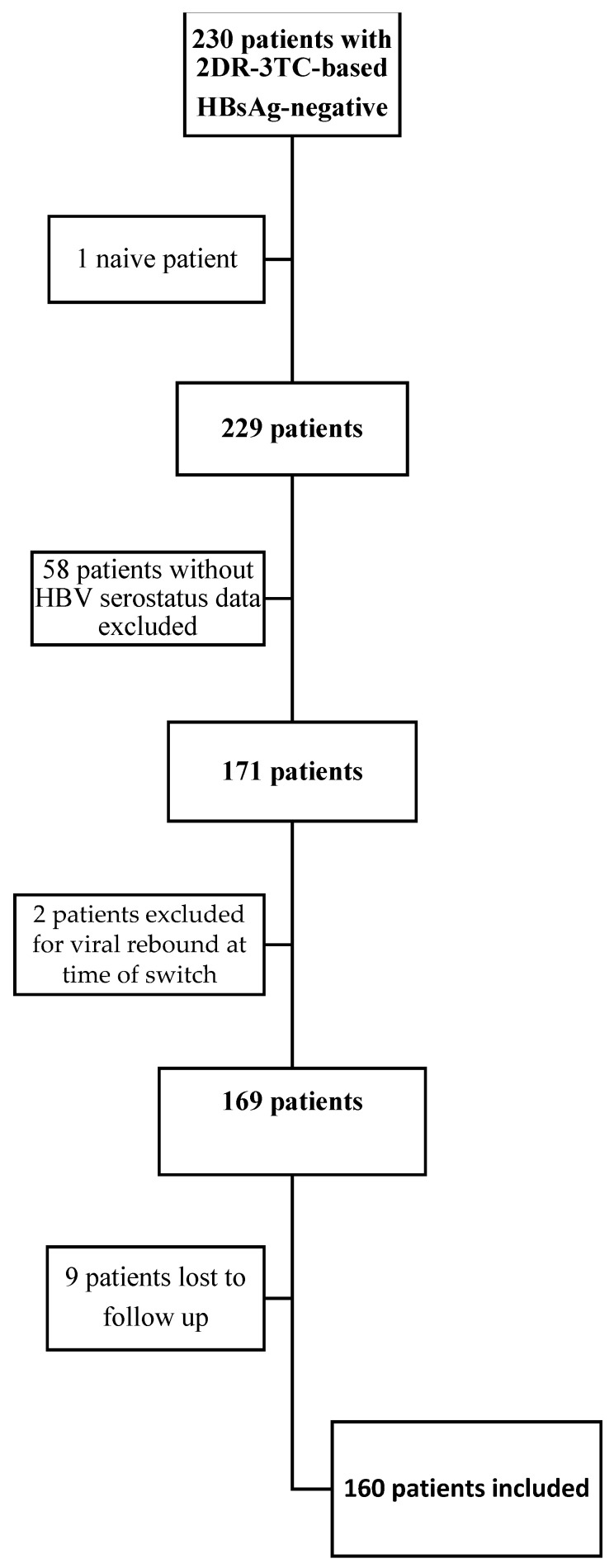
Patients inclusion and exclusion algorithm.

**Figure 2 viruses-15-00193-f002:**
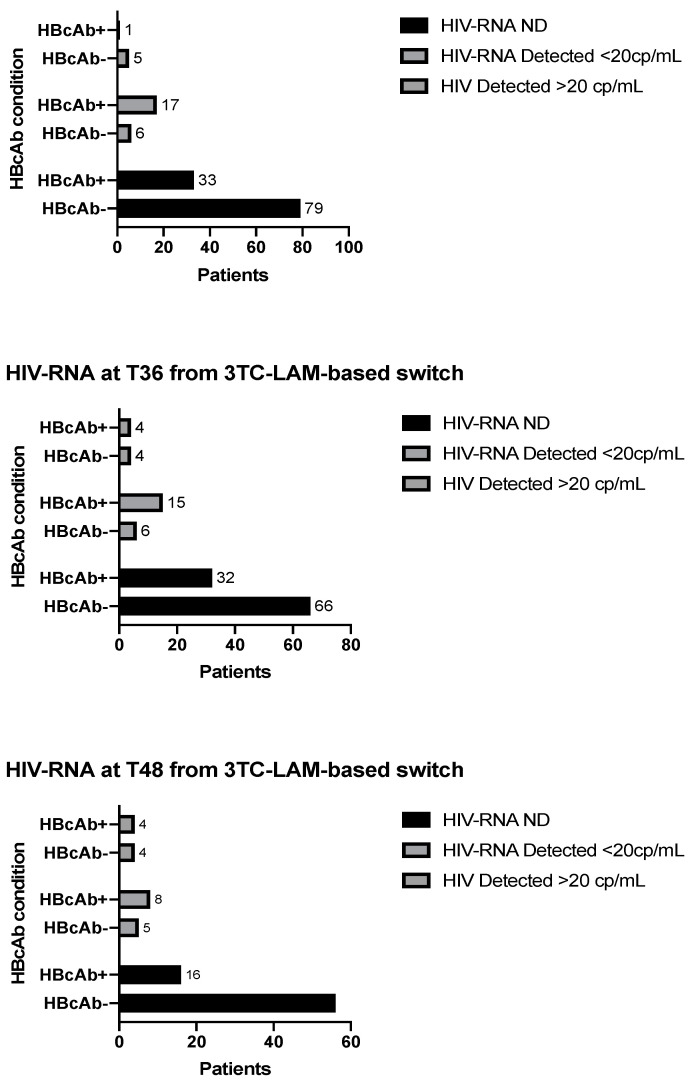
Analysis of primary and secondary endpoints at times 24, 36, and 48 for HBcAb-positive vs. HBcAb-negative patients.

**Table 1 viruses-15-00193-t001:** Characteristics of the study population.

	*n* = 160
Sex ratio M/F, %F	126/34 (21.2%)
Age, years	42.7 (32–54)
Calendar year of HIV diagnosis, median (IQR)	2013 (2008–2016)
Nadir CD4+, cell/mmc, median (IQR)	287 (123–414)
cART pre-switch, *n* (%):	
2NRTI + PI	48 (30.1%)
2NRTI + NNRTI	51 (32.1%)
2NRTI + INI	47 (29.6%)
Other	13 (8.2%)
cART TAF/TDF-containing pre-switch, *n* (%):	141 (88.1%)
CD4+ at 2DR switch, cell/mmc	753 (547–926)
HCVAb+ serology, *n* (%)	19 (11.9%)
Previous HIV viral rebound, *n* (%)	22 (13.8%)
HIV RNA 24 months pre-2DR switch, *n* (%):	
Undetectable	100 (62.5%)
<20 cp/mL Target detected	45 (28.1%)
Detectable >20 cp/mL	15 (9.4%)
HIV RNA 12 months pre-2DR switch, *n* (%):	
Undetectable	127 (79.4%)
<20 cp/mL Target detected	16 (10%)
Detectable >20 cp/mL	17 (10.6%)
HIV RNA at 2DR switch, *n* (%):	
Undetectable,	116 (72.5%)
<20 cp/mL Target detected	30 (18.7%)
Detectable >20 cp/mL	14 (8.8%)
2DR strategy, *n* (%):	
3TC + PI	37 (23.2%)
3TC + DTG	123 (76.9%)
HIV RNA 24 months post-2DR switch, *n* (%) *:	
Undetectable	112 (79.4%)
<20 cp/mL Target detected	23 (16.3%)
Detectable >20 cp/mL	6 (4.3%)
HIV RNA 36 months post-2DR switch, *n* (%) **:	
Undetectable	98 (77.2%)
<20 cp/mL Target detected	21 (16.6%)
Detectable >20 cp/mL	27 (6.2%)
HIV RNA 48 months post-2DR switch, *n* (%) ***:	
Undetectable	72 (77.4%)
<20 cp/mL Target detected	13 (14%)
Detectable >20 cp/mL	8 (8.6%)
AntiHBc+, *n* (%):	51 (31.9%)

*: Data available for 141 pts; **: Data available for 127 pts; ***: data available for 93 pts.

**Table 2 viruses-15-00193-t002:** Comparison of AntiHBc + Vs. AntiHBc − subpopulations.

	HBcAb+ (*n* = 51)	HBcAb− (*n* = 109)	*p*-Value
Sex ratio M/F, %F	39/12 (23.5%)	87/22 (20.2%)	0.62
Age, years, median (IQR)	47 (37–55)	40 (31–50)	**0.0014**
Calendar year of HIV diagnosis, median (IQR)	2007 (2000–2012)	2010 (2006–2017)	**0.039**
Nadir CD4+, cell/mmc, median (IQR)	222 (69–335)	266 (158–445)	**0.023**
History of HIV viral rebound, *n* (%)	10 (19.6%)	12 (11%)	0.12
cART pre-switch, *n* (%):			0.33
2NRTI + PI	16 (31.4%)	33 (30.3%)	
2NRTI + NNRTI	15 (29.4%)	36 (33%)	
2NRTI + INI	13 (25.5%)	34 (31.2%)	
Other	7 (13.7%)	6 (5.5%)	
cART TAF/TDF-containing pre-switch, *n* (%):	44 (86.3%)	97 (88.9%)	0.83
CD4+ at 2DR switch, cell/mmc, median (IQR)	683 (547–944)	680 (545–926)	0.91
HCVAb+ serology, *n* (%)	8 (15.7%)	11 (10.1%)	0.22
HIV RNA 24 months pre-2DR switch, *n* (%):			0.41
Undetectable	33 (64.7%)	67 (61.4%)	
<20 cp/mL Target detected	16 (31.3%)	29 (26.6%)	
Detectable >20 cp/mL	2 (4%)	13 (12%)	
HIV RNA 12 months pre-2DR switch, *n* (%):			0.40
Undetectable	38 (74.5%)	89 (81.6%)	
<20 cp/mL Target detected	6 (11.7%)	10 (9.2%)	
Detectable >20 cp/mL	7 (13.8%)	10 (9.2%)	
HIV RNA at 2DR switch, *n* (%):			0.87
Undetectable, *n* (%)	34 (66.8%)	83 (76.2%)	
<20 cp/mL Target detected	14 (27.5%)	20 (18.3%)	
Detectable >20 cp/mL	3 (5.9%)	6 (5.5%)	
Persistently TND pre-switch, *n* (%)	15 (29.4%)	43 (39.4%)	0.29
2DR strategy:			0.74
3TC + PI	11 (21.6%)	26 (23.9%)	
3TC + DTG	40 (78.4%)	83 (76.1%)	
HIV RNA 24 months post-2DR switch, *n* (%):			**<0.0001**
Undetectable	33 (64.7%)	79 (87.8%)	
<20 cp/mL Target detected	17 (33.3%)	6 (6.7%)	
Detectable >20 cp/mL	1 (2%)	5 (5.5%)	
HIV RNA 36 months post-2DR switch, *n* (%) *:			**0.011**
Undetectable	32 (62.7%)	66 (86.8%)	
<20 cp/mL Target detected	15 (29.4%)	6 (7.9%)	
Detectable >20 cp/mL	4 (7.9%)	4 (5.3%)	
HIV RNA 48 months post-2DR switch, *n* (%) **:			**0.021**
Undetectable	16 (57.2%)	56 (86.1%)	
<20 cp/mL Target detected	8 (28.6%)	5 (7.7%)	
Detectable >20 cp/mL	4 (14.2%)	4 (6.2%)	
Persistently TND during 48 months after switch, *n* (%) ***	1 (3.5%)	33 (50.7%)	**<0.0001**

*: Data available for 141 pts; **: Data available for 127 pts; ***: data available for 93 pts.

**Table 3 viruses-15-00193-t003:** Univariate and Multivariate analysis for estimated factors with predictive impact on Detected HIV-RNA during the entire follow-up 48 months after the 2DR-3TC-containing switch.

	*Univariable*		*Multivariable*	
	OR (95% CI)	*p*-Value	OR (95%)	*p*-Value
Age,	1.01 (0.94–1.04)	0.39	0.97 (0.96–1.03)	0.85
Calendar year of HIV infection	1.10 (0.98–1.19)	0.76	0.95 (0.89–1.02)	0.17
Nadir CD4+ cell count, >250/mm^3^	1 (0.99–1.12)	0.23	0.97 (0.63–1.51)	0.61
Previous HIV VF	3.04 (1.05–13.16)	0.028	2.00 (0.40–6.32)	0.39
HbcAb-positivity	9.32 (4.04–22.48)	0.003	7.46 (2.35–14.77)	**0.004**

## Data Availability

Data available on request from the corresponding author. The data are not publicy available due to privacy restrictions.

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
