# Peer review of "Role of HBcAb Positivity in Increase of HIV-RNA Detectability after Switching to a Two-Drug Regimen Lamivudine-Based (2DR-3TC-Based) Treatment: Months 48 Results of a Multicenter Italian Cohort"

_viruses, 2023, doi:10.3390/v15010193_

Round 1

Reviewer 1 Report

The article reflects an interesting aspect that is not usually taken into account in routine clinical practice but that could have long-term clinical repercussions in certain patients.

The study has some aspects that could overshadow the final results:

a) The lack of data on the backbone prior to the switch that could have conditioned the preliminary data considering that a therapy that includes tenofovir would better cover any HBV rebound. In this sense, as there is no difference in viral loads from baseline, it would be interesting to know if the patients in Hbc Ab were on TDF-containing therapies before switching.

b) Lack of data on HBV DNA. Having HBc Ab without having DNA makes it impossible to know if it is the virus itself that is favoring the rebound of HIV or it may be influenced by other external factors such as degree of fibrosis, active HCV coinfection (it must be addressed with PCR+ and include the serology percentages in the table 2 because they do not appear), adherence could interfere with the final results.

In the results, the authors have described the study population and most of these data are duplicated in tables. I would shorten the text and refer to the tables. In general, I would shorten the results regarding the main characteristics of the population and include them in the tables.

In Table 1 the authors use the term detectable, and in Table 2 > 20 copies/ml is used. In my opinion, I would use the latter option as in clinical practice detectable or undetectable are mostly associated with 50 copies/ml and should avoid confusing readers. In this sense, I would always try to use low-level viremia above > 30 copies/ml.

It will be interesting to have information about the main features of the DTG/3TC. Perhaps a supplementary table it should be included.

Under discussion, lines 216-117 I would not repeat the numerical results. I would limit to describe the main findings and how they do or do not relate to the published data.

Another aspect that needs to be addressed is to define the possible clinical implications of the persistence of this low level of viremia in the follow-up of patients with 2DR-3TC-based regimens because till the moment this aspects are not taken into account by clinicians.

Reviewer 2 Report

The paper looks into the impact of HBV on the course of HIV infection. Previous work from others suggests no impact, however the authors have published on the same topic over recent years finding an association between HBcAb positivity and adverse HIV virological and immunological outcomes.

In this paper they refer to 48m virological outcomes from a retrospective cohort study. 

A number of things about the study are a little unclear to me from the manuscript in its current state -

1. Methodology - were these data prospectively collected on a previous retrospective cohorts?

unclear what lost to follow up means here?

2. Terminology - missing definition and use of the term 2DR-3TC based - this is defined in the abstract but I'm not sure about during the text?

Table 1 uses '2DR LAM-containing' ? (prev 3TC)

3. Information - presumably the HBsAg status of all participants was negative throughout? (I could not see this mentioned), 

- the conclusion talks about TAF/TDF but not clear how many were on tenofovir pre-switch

4. significance of the outcome - is there evidence that detectable<20 is significantly different to non-detectable?

5. any information on M184V mutations?

6. discussion - may be worth including some information about clinical outcomes?

Round 2

Reviewer 1 Report

I acept the amnuscript for publication after the review provided